

# A new metric for climate models that includes field and spatial dependencies using Gaussian Markov Random Fields

A. Nosedal-Sanchez[1,2], C. S. Jackson[3], and G. Huerta[1]

[1]Department of Mathematics and Statistics, The University of New Mexico, Albuquerque, USA
[2]Department of Mathematical and Computational Sciences, University of Toronto, Mississauga, Canada
[3]Institute for Geophysics, The University of Texas at Austin, Austin, USA

Received: 15 November 2015 – Accepted: 4 December 2015 – Published: 15 January 2016

Correspondence to: C. S. Jackson (charles@ig.utexas.edu)

Published by Copernicus Publications on behalf of the European Geosciences Union.

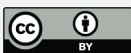

**Abstract**

A new metric for climate model evaluation has been developed that potentially mitigates some of the limitations that exist for observing and representing field and space dependencies of climate phenomena. Traditionally such dependencies have been ignored when climate models have been evaluated against observational data, which makes it difficult to assess whether any given model is simulating observed climate for the right reasons. The new metric uses Gaussian Markov Random Fields for estimating field and space dependencies within a first order grid point neighborhood structure. We illustrate the ability of Gaussian Markov Random Fields to represent empirical estimates of field and space covariances using "witch hat" graphs. We further use the new metric to evaluate the tropical response of a climate model (CAM3.1) to changes in two parameters important to its representation of cloud and precipitation physics. Overall, the inclusion of dependency information did not alter significantly the recognition of those regions of parameter space that best approximated observations. However there were some qualitative differences in the shape of the response surface that suggest how such a measure could affect estimates of model uncertainty.

# 1 Introduction

Within the climate assessment community, there is an interest to develop metrics of how well simulations reproduce observed climate for purposes of comparing models, driving model development, and evaluating model prediction uncertainties (Gleckler et al., 2008; Reichler and Kim, 2008; Santer et al., 2009; Knutti et al., 2010; Weigel et al., 2010). Nevertheless, a certain level of skepticism exists about whether a scalar metric can be sufficiently informative for these purposes. Climate phenomena involve interactions of multiple quantities on a wide range of time and space scales from minutes to decades (and longer) and from meters to planetary scales. Thus it can be challenging to summarize what is physically meaningful. The most common approach

Discussion Paper | Discussion Paper | Discussion Paper | Discussion Paper |

**GMDD**

doi:10.5194/gmd-2015-250

**A new metric for climate models**

A. Nosedal-Sanchez et al.

Title Page

Abstract | Introduction

Conclusions | References

Tables | Figures

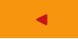 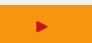

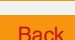 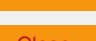

to climate model evaluation among climate scientists is to display maps of long-term means of well-known quantities (e.g. temperature, sea-level pressure, precipitation) whose distribution is familiar and well understood in order to identify sources of model error. The Taylor metric that is often generated as part of model evaluation is based on spatial means of squared grid point errors for individual quantities (Taylor, 2001). Such measures neglect field and space dependencies and thus may be insensitive to mechanisms giving rise to model errors. There is a need to develop metrics that can evaluate whether a model is capturing observed space and field relationships sufficiently well (Braverman et al., 2011). The hope is that by accounting for relationship information within climate model metrics, they will prove to be more useful for scientific evaluation.

Given that there is only a limited amount of observations available to quantify field and space relationships of climate phenomena, data assimilation is the most common approach to fill in gaps in the observational record of a climate model (Trenberth et al., 2008). While assimilation data products help solve some aspects of the problem of how one compares point measurements to the scales resolved by climate models, these data products include the space and field dependencies of the model that was used to assimilate the data. Here we introduce a new kind of metric based on Gaussian Markov Random Fields that only needs limited data to decipher space and field dependencies of climate phenomena.

We define a new $Z$ test statistic, alternatively referred to as a log-likelihood or cost for assessing the significance of a discrepancy between model output and observations. The statistic makes use of Gaussian Markov Random Fields to estimate field and space dependencies that exist within gridded climate model output that can be assessed against space and field dependent observational data. The matrix form of the test statistic is given by:

$$\boldsymbol{v}^{\top}\mathbf{S}^{-1} \otimes (\alpha\mathbf{I} + (1-\alpha)\mathbf{Q})\boldsymbol{v} \tag{1}$$

**GMDD**

doi:10.5194/gmd-2015-250

**A new metric for climate models**

A. Nosedal-Sanchez et al.

Discussion Paper | Discussion Paper | Discussion Paper | Discussion Paper

**GMDD**

doi:10.5194/gmd-2015-250

**A new metric for climate models**

A. Nosedal-Sanchez et al.

where $\boldsymbol{v}$ is the vector of differences between model output and observations with a length given by the product of the number of observational fields and number of grid points, $n_{\text{obs}} \times n_{\text{pts}}$, $\alpha$ is a scalar with a value close to zero, $\mathbf{I}$ stands for an identity matrix (a diagonal matrix of ones) of a dimension corresponding to $\boldsymbol{v}$, and $\mathbf{Q}$ is a precision matrix of dimension $n_{\text{pts}} \times n_{\text{pts}}$ from a Gaussian Markov Random Field (GMRF) induced by a first order neighborhood structure. This cost function captures field dependencies through $\mathbf{S}^{-1}$ which is a matrix of dimension $n_{\text{obs}} \times n_{\text{obs}}$ where each of its elements represents a spatial-average of grid point variances and covariances between fields. The spatial dependency between grids is approximated through $\mathbf{Q}$. The quantity $\alpha$ could be interpreted as a weight of the spatial relationship between grid cells. The Kronecker product $\otimes$ provides a means for associating the different matrix dimensions of the metric, essentially combining its field and space components.

The sections of this paper explain, test, and provide examples of how various components of Eq. (1) work. Section 2 gives a brief introduction to GMRFs. This section will allow us to understand how $\mathbf{Q}$ is obtained and the information that it provides about spatial dependency between grid cells. In this section we also define and discuss Kronecker products, and how to use this concept to generalize GMRF ideas to deal with more than one field. Section 3 introduces a graph for testing the extent to which Eq. (1) captures observed variance-covariances of tropical temperature, precipitation, sea level pressure, and upper level winds. Finally, in Sect. 4, we consider the field and space dependencies that are captured by the GMRF-based metric within the response of an atmospheric general circulation model CAM3.1 to two model parameters important to cloud and precipitation physics. What we learned in general is that including the space and field dependencies provides some qualitatively different perspectives about which model configurations are more similar to what is observed. For the example we consider, the effects of space dependencies turn out to be more critical than field dependencies.

## 2  Gaussian Markov Random Fields (GMRFs)

A Gaussian Markov Random Field (GMRF) is a special case of a multivariate normal distribution, one that satisfies additional properties related to conditional independence. The density of a normal random vector $\boldsymbol{x} = (x_1, x_2, ..., x_n)^\mathsf{T}$ (where "T" denotes the operation of transposing a column to a row), with mean $\boldsymbol{\mu}$ ($n \times 1$ vector) and covariance matrix $\boldsymbol{\Sigma}$ ($n \times n$ matrix), is

$$f(\boldsymbol{x}) = (2\pi)^{-n/2} |\boldsymbol{\Sigma}|^{-\frac{1}{2}} \exp\left\{ -\frac{1}{2}(\boldsymbol{x} - \mu)^\mathsf{T}\boldsymbol{\Sigma}^{-1}(\boldsymbol{x} - \mu) \right\} \tag{2}$$

Here, $\mu_i = E(x_i)$, $\Sigma_{ij} = \text{Cov}(x_i, x_j)$, and $\Sigma_{ii} = \text{Var}(x_i) > 0$. All eigenvalues of $\boldsymbol{\Sigma}$ must be greater than zero, otherwise $\boldsymbol{\Sigma}$ becomes a singular matrix and does not define a valid multivariate normal distribution. It can also be shown that if all eigenvalues of $\boldsymbol{\Sigma}$ are positive then all eigenvalues of $\boldsymbol{\Sigma}^{-1}$ are also greater than zero. We define $\mathbf{Q} = \boldsymbol{\Sigma}^{-1}$ and refer to $\mathbf{Q}$ as the precision matrix, and denote $\boldsymbol{x} \sim \mathbf{N}(\boldsymbol{\mu}, \mathbf{Q})$ to represent $\boldsymbol{x}$ as a multivariate normal distribution with vector mean $\boldsymbol{\mu}$ and precision matrix $\mathbf{Q}$.

### 2.1  Precision matrix of a GMRF

The precision matrix $\mathbf{Q}$ of a GMRF is an operator for obtaining information about dependencies among neighboring grid cells. Although $\mathbf{Q}$ is sparse, its inverse, as a model for the covariance matrix $\boldsymbol{\Sigma}$, presumes all grid points are conditionally dependent. $\mathbf{Q}$ needs to be constructed such that it:

- Reflects the kind of spatial dependency we assume our data has.

- Yields a legitimate covariance matrix, $\boldsymbol{\Sigma}$, i.e. symmetric and positive definite, so that it can be used to compute a likelihood function.

Consider $\boldsymbol{x}$, a vector of measurements on a $2 \times 2$ lattice, as represented in Fig. 1. Assume a neighborhood structure between the four elements of $\boldsymbol{x}$. In Fig. 2, the neighbors for each element of $\boldsymbol{x}$ are defined graphically. Given the neighborhood structure

Discussion Paper | Discussion Paper | Discussion Paper | Discussion Paper | Discussion Paper |

**GMDD**

doi:10.5194/gmd-2015-250

**A new metric for climate models**

A. Nosedal-Sanchez et al.

shown in Fig. 2, the precision matrix that works for this problem is

$$\mathbf{Q} = \begin{pmatrix} 2 & -1 & -1 & 0 \\ -1 & 2 & 0 & -1 \\ -1 & 0 & 2 & -1 \\ 0 & -1 & -1 & 2 \end{pmatrix}$$

which follows these rules,

- $\mathbf{Q}_{ij} = -1$, if $x_i$ and $x_j$ are neighbors.
- $\mathbf{Q}_{ij} = 0$, if $x_i$ and $x_j$ are not neighbors.
- $\mathbf{Q}_{ii}$ gives the total number of neighbors of $x_i$.

While the implementation of GMRFs is simple, the theory and mathematics are rather involved. A fuller description of the mathematics of this example is provided in the supplemental material. It may also not be immediately clear to a physical scientist that such a simple specification, where only relationships among neighboring grid cells are taken into account, would be sufficient to quantify correlated quantities across large distances. The mathematics of working with precisions allows one to infer the net effect of long distance relationships through relationship information that exists among neighboring cells. While the GMRF approach does not include information about particular teleconnection structures such as ENSO, the approach is sensitive to how changes in large scale conditions induce local covariances across multiple fields within the entire domain. In this way teleconnections are represented through a conditional dependence.

A problem arises in that one of the eigenvalues of the $\mathbf{Q}$ matrix is 0, which implies that this definition of the precision matrix does not induce an invertible covariance matrix. This problem is solved by using $\alpha \mathbf{I} + (1 - \alpha)\mathbf{Q}$, instead of $\mathbf{Q}$. If $\alpha$ is small, the neighborhood structure remains essentially unchanged. Section 3 describes our approach to specifying a value for $\alpha$.

**GMDD**

doi:10.5194/gmd-2015-250

**A new metric for climate models**

A. Nosedal-Sanchez et al.

Discussion Paper | Discussion Paper | Discussion Paper | Discussion Paper |

## 2.2 Generalizing concepts to deal with multiple fields

The generalization of $\mathbf{Q}$ to handle multiple fields will be illustrated by an example using $\boldsymbol{x}$ and $\boldsymbol{y}$ which represent observations for two different fields of interest. These observations are taken on a $2 \times 2$ lattice. First, $\boldsymbol{x}$ and $\boldsymbol{y}$ are combined to form one vector $\boldsymbol{v}$ as follows: $\boldsymbol{v}^{\mathsf{T}} = (x_1, x_2, x_3, x_4, y_1, y_2, y_3, y_4)$. The average covariances among these observations can be represented by a $2 \times 2$ matrix between the first field, $\boldsymbol{x}$, and the second field, $\boldsymbol{y}$:

$$\mathbf{S} = \begin{pmatrix} \sigma_{11} & \sigma_{12} \\ \sigma_{12} & \sigma_{22} \end{pmatrix}$$

where $\mathrm{Var}(\boldsymbol{x}) = \sigma_{11}$, $\mathrm{Var}(\boldsymbol{y}) = \sigma_{22}$, and $\mathrm{Cov}(\boldsymbol{x}, \boldsymbol{y}) = \sigma_{12}$. Recalling that the correlation between fields 1 and 2 is defined as: $\rho = \frac{\sigma_{12}}{\sqrt{\sigma_{11}\sigma_{22}}}$, one may show that the inverse of $\mathbf{S}$ is

$$\mathbf{S}^{-1} = \begin{pmatrix} \frac{1}{\sigma_{11}(1-\rho^2)} & \frac{-\rho}{(1-\rho^2)\sqrt{\sigma_{11}\sigma_{22}}} \\ \frac{-\rho}{(1-\rho^2)\sqrt{\sigma_{11}\sigma_{22}}} & \frac{1}{\sigma_{11}(1-\rho^2)} \end{pmatrix} =: \begin{pmatrix} S_{11}^{-1} & S_{12}^{-1} \\ S_{12}^{-1} & S_{22}^{-1}. \end{pmatrix}$$

If we consider the Kronecker product in Eq. (1) when $\alpha = 0$,

$$\mathbf{S}^{-1} \otimes \mathbf{Q} = \begin{pmatrix} S_{11}^{-1}\mathbf{Q} & S_{12}^{-1}\mathbf{Q} \\ S_{21}^{-1}\mathbf{Q} & S_{22}^{-1}\mathbf{Q} \end{pmatrix}$$

then

$$\boldsymbol{v}^{\mathsf{T}}\mathbf{S}^{-1} \otimes \mathbf{Q}\boldsymbol{v} = S_{11}^{-1}\boldsymbol{x}^{\mathsf{T}}\mathbf{Q}\boldsymbol{x} + S_{12}^{-1}\boldsymbol{y}^{\mathsf{T}}\mathbf{Q}\boldsymbol{x} + S_{21}^{-1}\boldsymbol{x}^{\mathsf{T}}\mathbf{Q}\boldsymbol{y} + S_{22}^{-1}\boldsymbol{y}^{\mathsf{T}}\mathbf{Q}\boldsymbol{y}.$$

In this last expression, one can see that the inverse of $\mathbf{S}$ in combination with the Kronecker product with $\mathbf{Q}$ includes terms involving cross products between fields. The supplemental carries this expression one step further by estimating the conditional mean for the the first element of $\boldsymbol{v}$ to illustrate how this element is related to itself and its neighbors across multiple fields.

Discussion Paper | Discussion Paper | Discussion Paper | Discussion Paper |

**GMDD**

doi:10.5194/gmd-2015-250

**A new metric for climate models**

A. Nosedal-Sanchez et al.

Discussion Paper | Discussion Paper | Discussion Paper | Discussion Paper |

**GMDD**

doi:10.5194/gmd-2015-250

**A new metric for climate models**

A. Nosedal-Sanchez et al.

## 3 A test of GMRF estimates of variance

GMRFs provide a way to approximate field and space dependencies contained in the inverse covariance matrix $\Sigma^{-1}$ of Eq. (1) by its GMRF equivalent $S^{-1} \otimes (\alpha I + (1-\alpha)Q)$. In this section, we will test how well GMRFs are able to reproduce observed space
and field dependencies. This may be achieved by comparing field and spatial variance and covariance estimates obtained from the inverse of the GMRF equation with those obtained empirically from observational data. It turns out this comparison is sensitive to the value that is selected for $\alpha$. Fortunately, the optimal choice of $\alpha$ depends only on geometric considerations of the neighborhood model that is used for GMRF and
the number of grid points in the fields and not the properties of the field data. We introduce a "witch hat" graph that provides a compact summary of variance-covariance information between these two methods in order to show that GMRFs do a reasonable job approximating observed field and space relationships.

### 3.1 Finding an appropriate value of $\alpha$

In the effort to compare space and field dependencies approximated by GMRFs with empirical estimates we need to determine an optimal value for $\alpha$. In order to carry out this comparison, we need to find the inverse of $S^{-1} \otimes (\alpha I + (1-\alpha)Q)$, our proposed precision matrix based on GMRF. Using results of Kronecker products, we have that $\left[ S^{-1} \otimes (\alpha I + (1-\alpha)Q) \right]^{-1} = S \otimes (\alpha I + (1-\alpha)Q)^{-1}$. Letting $Q^* = (\alpha I + (1-\alpha)Q)^{-1}$, then $S \otimes$
$Q^*$ for two fields can be written as

$$\begin{pmatrix} S_{11}Q^* & S_{12}Q^* \\ S_{12}Q^* & S_{22}Q^* \end{pmatrix}.$$

If $n$ is the total number of grid points of the lattice, $S \otimes Q^*$ is a $(2 \times n) \times (2 \times n)$ covariance matrix. Note that each element of $\text{diag}(S_{ij}Q^*)$ contains the estimated variance or covariance at each grid point for fields $i$ and $j$ using a GMRF where $i$ can be equal

to $j$. If we average these estimates across the whole lattice, we obtain $G_{ij}$, the GMRF estimate of the variance or covariance. Therefore,

$$G_{ij} = \frac{S_{ij} \sum_{k=1}^{n} Q^*_{kk}}{n} = \frac{S_{ij} tr(\mathbf{Q}^*)}{n} \qquad (3)$$

where $tr(\mathbf{Q}^*)$ denotes the trace of $\mathbf{Q}^*$ and $Q^*_{kk}$ are its diagonal elements. We will now select a value for $\alpha$ that allows the GMRF estimate for field variances and covariances to be equal, on average, to what has been calculated for $\mathbf{S}$. In order to achieve this, $G_{ij}$ needs to equal $S_{ij}$. Satisfying this condition is equivalent to finding the solution for

$$\frac{tr(\mathbf{Q}^*)}{n} = 1. \qquad (4)$$

It may not be so obvious what the diagonal elements of $\mathbf{Q}^*$ are. However, one can use the fact that for any matrix $\mathbf{A}$ that admits a Singular Value Decomposition, $tr(\mathbf{A})$ is equal to sum of its eigenvalues. In our case, if the eigenvalues of $\mathbf{Q}$ are $\lambda_1, \lambda_2, \ldots, \lambda_n$, the eigenvalues of $\alpha \mathbf{I} + (1 - \alpha)\mathbf{Q}$ are $\alpha + (1 - \alpha)\lambda_1, \alpha + (1 - \alpha)\lambda_2, \ldots, \alpha + (1 - \alpha)\lambda_n$. The eigenvalues of $\mathbf{Q}^* = (\alpha \mathbf{I} + (1 - \alpha)\mathbf{Q})^{-1}$ are $(\alpha + (1 - \alpha)\lambda_1)^{-1}, (\alpha + (1 - \alpha)\lambda_2)^{-1}, \ldots, (\alpha + (1 - \alpha)\lambda_n)^{-1}$. This implies that in order to satisfy Eq. (4), we need to find $\alpha$ from

$$f(\alpha) = \sum_{i=1}^{n} \frac{1}{n(\alpha + (1 - \alpha)\lambda_i)} = 1. \qquad (5)$$

Figure 3 shows the relationship between various values of $\alpha$ and $f(\alpha)$. The eigenvalues used to obtain this figure correspond to a precision matrix, $\mathbf{Q}$, for a GMRF induced by a first order neighborhood structure and considering a $128 \times 22$ lattice (which is the dimension of our data). From the figure we can see that the curve crosses the value of 1 when $\alpha$ is close to 0. By using linear interpolation, we determine that $\alpha$ is approximately 0.0026.

Discussion Paper | Discussion Paper | Discussion Paper | Discussion Paper

**GMDD**

doi:10.5194/gmd-2015-250

**A new metric for climate models**

A. Nosedal-Sanchez et al.

Discussion Paper | Discussion Paper | Discussion Paper | Discussion Paper |

**GMDD**

doi:10.5194/gmd-2015-250

**A new metric for climate models**

A. Nosedal-Sanchez et al.

## 3.2 "Witch hat" comparison test

To illustrate any differences that may exist between empirical estimates of the covariance matrix $\mathbf{\Sigma}$ and its GMRF equivalent $\mathbf{S} \otimes (\alpha\mathbf{I} + (1 - \alpha)\mathbf{Q})^{-1}$, we rely on a graph that shows the spatial average grid point variance and covariances as a function of distance for cells and their neighbors. We compute the average entries of the covariance matrix corresponding to each grid cell and the corresponding element to the north (for the positive distances) or to the south (for the negative distances) relative to the main diagonal of the matrix. The zero distance case is the average of variances of the main diagonal. Alternatively, we can produce a graph that considers the east and west directions. On average, covariances decrease with distance making the graph have the shape of a witch's hat. This graph is symmetric because covariance matrices are symmetric.

Figure 4 shows a witch hat test of estimated variances for air temperatures simulated by the Community Atmosphere Model version 3.1 (CAM3.1). The variances are estimated from 15 samples of two year mean summertime temperatures. Setting $\alpha = 1$ provides a solution to Eq. (5), however, this will shut down the effect of $\mathbf{Q}$ and only the variances at the reference point (lag 0) will be well estimated. On the other hand, when $\alpha = 0.0026$, we allow $\mathbf{Q}$ to play more of a role which results in a better representation of covariances at neighboring points (lags different of zero).

## 4 Climate response to uncertain parameters

In this section we show how inclusion of field and space dependencies using GMRF affect comparisons of the Community Atmosphere Model (CAM3.1) (Collins et al., 2006) with observations. We consider CAM3.1's response to to changes in parameter *ke*, which controls rain drop evaporation rates, and parameter *c0*, which controls precipitation efficiency through conversion of cloud water to rain water. For this comparison we only consider the response for the June, July, and August (JJA) seasonal mean between 30° S to 30° N on four variables including 2 m air temperature (TREFHT), 200-

mbar zonal winds ($U$), sea level pressure (PSL), and precipitation (PRECT). Experiments with CAM3.1 use observed climatological sea surface temperatures and sea ice extents. Each experiment with CAM3.1 is 32-years in duration.

The observational data that is used to evaluate the model comes from a reanalysis product ECMWF-ERA interim (Uppala et al., 2005) for 2 m air temperature, 200-mbar zonal winds, and sea level pressure and GPCP (Adler et al., 2009) for precipitation. We make use of approximately 30 years of JJA mean fields between 1979 and 2009. For constructing **S**, we calculate variances from 2-year means (i.e. 15 samples).

A total of 64 experiments were completed, varying each of the two parameters within an 8 × 8 lattice. For each experiment we calculate three versions of GMRF-based cost (Eq. 1). The first version is the traditional cost based on the assumption of space and field independence set here by setting the off diagonal components of **S** to zero and setting $\alpha = 1$ . This approach is similar to what has been done previously for Taylor (2001). The second version of evaluating the cost takes field dependencies into account by including all components of **S** and setting $\alpha = 1$. The third version for the cost takes field and space dependencies into account by including all components of **S** and setting $\alpha = 0.0026$.

The correlation matrix, **R**, corresponding to the **S** matrix of 2-year JJA seasonal mean variances and covariances, as estimated from 30 years of observations, is described in Table 1.

The primary field correlations are the values of ($-0.313$) and ($-0.219$) occurring between sea level pressure (PSL) and 2 m air temperature (TREFHT), and precipitation (PRECT) and sea level pressure (PSL), respectively. These correlations make physical sense in that precipitation mainly occurs within low pressure storm systems which tends to cool the underlying surface. The other correlations are minimal and there is not a good physical argument supporting their relationship. Figure 5 shows a comparison of the three versions of the GMRF-based cost for the 64 experiments within an 8 × 8 lattice. All versions of cost result in qualitatively similar results with high and low cost values roughly in the same portions of parameter space. The main difference among the

**GMDD**

doi:10.5194/gmd-2015-250

**A new metric for climate models**

A. Nosedal-Sanchez et al.

Interactive Discussion

**GMDD**

doi:10.5194/gmd-2015-250

**A new metric for climate models**

A. Nosedal-Sanchez et al.

versions of cost comes from taking space dependencies into account within the field-space version. In this case, extremely low values of $ke$ result in higher metric values. Figure 6 examines the reasons for this by graphing the different field contributions to the GMRF-based costs for a slice where $c0 = 0.0035$ which corresponds to one of the rows

of the lattice. By plotting everything differenced from metric values at $ke = 3 \times 10^{-6}$, one can learn that the biggest qualitative difference comes from cost values associated with 2 m air temperature. Closer inspection of differences between model output and observations of 2 m air temperature (not shown) indicates that the traditional cost is likely reflecting large-scale differences over the Southern Hemisphere oceans. Inclusion of

space dependencies places much greater significance on smaller-scale anomalies occurring over the continents, particularly over the Andes Mountains. This finding is a result of the mathematics of GMRF. It does not imply that the large-scale errors are of lesser scientific importance. It only means that GMRF is less sensitive to large-scale anomalies, perhaps because they are associated with fewer degrees of freedom than

highly structured errors. Understanding whether and how these distinctions aid model assessment needs further study. We do find it reassuring that GMRF-based metrics of distance to observations are similar, at least in the example provided, to a traditional metric.

## 5 Summary

We have developed a new test statistic as a scalar measure of model skill or cost for evaluating the extent to which climate model output captures observed field and space relationships using Gaussian Markov Random Fields (GMRFs). The challenge has been that few observations exist for establishing a meaningful observational basis for quantifying field and space relationships of climate phenomena. Much of the data

that is typically used for model evaluation is suspected of having its own relationship biases introduced by the numerical model that is used to synthesize measurements into gridded products. The GMRF-based metric overcomes some of these limitations

by considering field and space variations within a neighborhood structure thereby lowering the metric's data requirements. The form of the metric separates space and field dependencies using a Kronecker product that, when multiplied out, has all the terms necessary to represent how different points in space are tied together across multiple
fields. We also include a scalar $\alpha$ that weights the importance of spatial relationships between grid cells. Its optimal value turns out to be independent of the data type which aids the use of GMRFs for comparing model output to data across multiple fields. Using "witch hat" graphs, we show a first order (nearest neighborhood) structure does an excellent job of capturing empirical estimates of field and space relationships. We
have applied three versions of cost that selectively turn on or off field and space dependencies in a climate model (CAM3.1) output against observational products for tropical JJA climatologies for 2 m air temperature, sea level pressure, precipitation, and 200-mbar zonal winds. The results show subtle, but potentially important differences among these versions of the cost which may prove beneficial for selecting models that capture
observed climate phenomena for the right reasons.

**Code and data availability**

R code and data for generating Figs. 5 and 6 can be obtained through https://zenodo.org/record/33765, Nosedal-Sanchez et al. (2015).

*Acknowledgements.* This material is based upon work supported by the U.S. Department of
20 Energy Office of Science, Biological and Environmental Research Regional & Global Climate Modeling Program under Award Numbers DE-SC0006985 and DE-SC0010843. Nosedal was partially supported by the National Council of Science and Technology of Mexico (CONACYT).

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



**Table 1.** Correlation matrix, **R**, corresponding to the **S** matrix of 2-year JJA seasonal mean variances and covariances, as estimated from 30 years of observations of precipitation (PRECT), sea level pressure (PSL), 2 m air temperature (TREFHT), and 200-mbar zonal winds (U).

|        | PRECT  | PSL    | TREFHT | U      |
|--------|--------|--------|--------|--------|
| PRECT  | 1      | −0.219 | −0.047 | 0.015  |
| PSL    | −0.219 | 1      | −0.313 | −0.112 |
| TREFHT | −0.047 | −0.313 | 1      | −0.145 |
| U      | 0.015  | −0.112 | −0.145 | 1      |

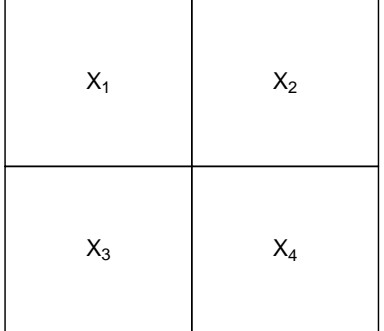

**Figure 1.** Graphical representation of 2 × 2 lattice and elements of *x*.

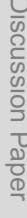

# GMDD

doi:10.5194/gmd-2015-250

**A new metric for climate models**

A. Nosedal-Sanchez et al.

Discussion Paper | Discussion Paper | Discussion Paper | Discussion Paper | Discussion Paper |

GMDD

doi:10.5194/gmd-2015-250

**A new metric for climate models**

A. Nosedal-Sanchez et al.

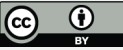

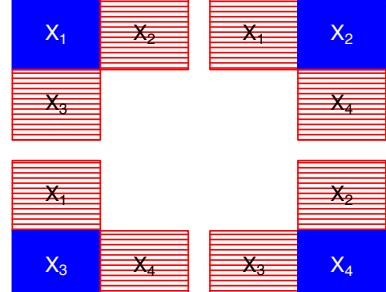

**Figure 2.** Neighbors of $x_1$, $x_2$, $x_3$ and $x_4$.

Discussion Paper | Discussion Paper | Discussion Paper | Discussion Paper | Discussion Paper |

**GMDD**

doi:10.5194/gmd-2015-250

**A new metric for climate models**

A. Nosedal-Sanchez et al.

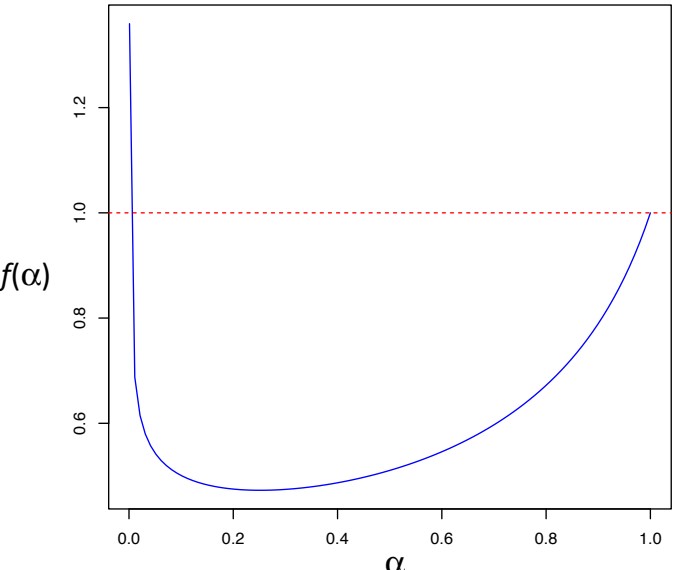

**Figure 3.** $\alpha$ vs. $f(\alpha)$.

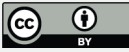

Discussion Paper | Discussion Paper | Discussion Paper | Discussion Paper | Discussion Paper |

**GMDD**

doi:10.5194/gmd-2015-250

**A new metric for climate models**

A. Nosedal-Sanchez et al.

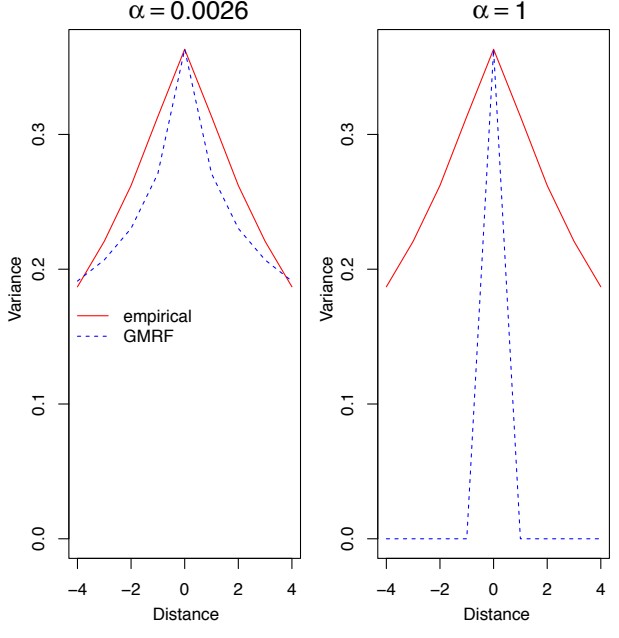

**Figure 4.** "Witch hat" graphs for air temperature on a 128 × 22 lattice of the tropics from 30° S to 30° N. The empirical estimates are given by the solid red line. The GMRF estimate is given by the dashed blue line.

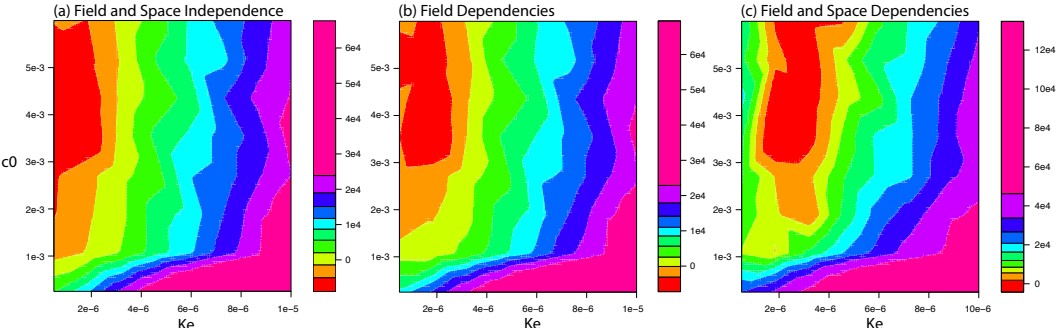

**Figure 5.** Three versions of the GMRF-based cost as a function of two CAM3.1 parameters *ke* and *c0* that assumes the data has **(a)** field and space independence, **(b)** field dependencies, and **(c)** field and space dependencies. Each color represents ten percentiles of the cost distribution. The cost is shown relative to the value of the default model configuration.

Discussion Paper | Discussion Paper | Discussion Paper | Discussion Paper

**GMDD**

doi:10.5194/gmd-2015-250

**A new metric for climate models**

A. Nosedal-Sanchez et al.

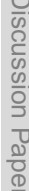

**GMDD**

doi:10.5194/gmd-2015-250

**A new metric for climate models**

A. Nosedal-Sanchez et al.

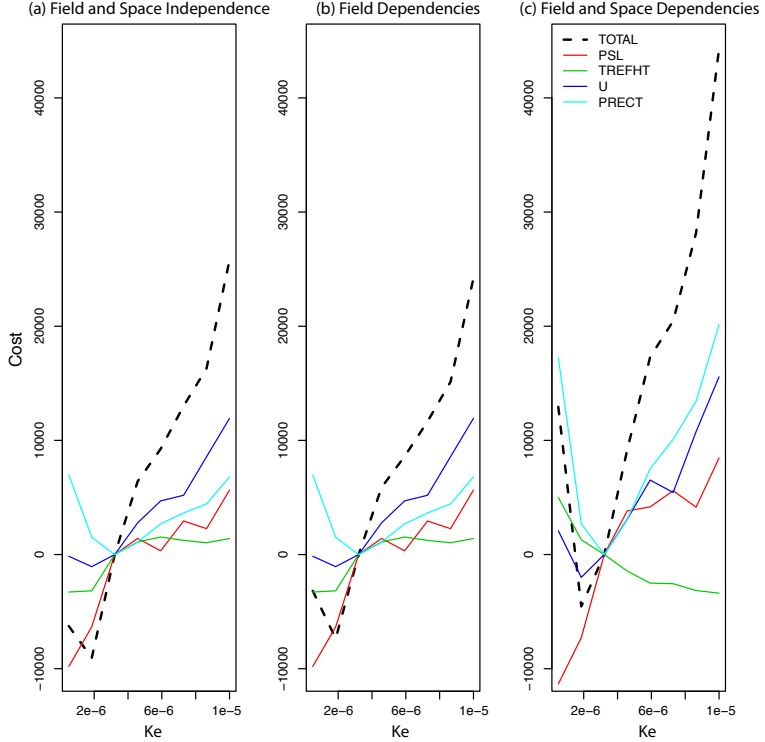

**Figure 6.** Different field contributions to the GMRF-based costs for a slice of Fig. 5 where $c0 = 0.0035$. Cost values are relative to the default parameter setting for $ke$. Note that total cost (black dashed line) is a weighted sum of field contributions as given by $\mathbf{S}^{-1}$ with contributions from sea level pressure (PSL, red line), 2-m air temperature (TREFHT, green line), 200-mbar zonal winds ($U$, blue line), and total precipitation (PRECT, cyan line).