# Peer review of "A new test statistic for climate models that includes field and spatial dependencies using Gaussian Markov Random Fields"

_Geoscientific Model Development, 2015_

## Referee Comment (RC1) · Anonymous Referee #1 · 26 Feb 2016

Summary Statement: Nosedal-Sanchez et al introduce Gaussian Markov random fields (GMRF) to account for co-variations across space and between fields while comparing climate model output to observations. These types of co-variations are potentially interesting to GMD readers and important to consider in evaluating climate model performance because assumptions of independence can lead to overconfidence and incorrect statistical inferences. While the presentation of the GMRF is reasonable and the illustration of the method to CAM3 data informative, there are areas where the manuscript can be improved, both in terms of readability and technical clarity. A number of items are listed below in random order. After addressing these items, some of which may require revisions, I should be in a position to recommend this manuscript

for publication in GMD.

Item 1. Conditional independence is an assumption underlying Markov random fields. For three variables A, B, and C, the joint probability distribution of A and B conditioned on C, written as p(A,B|C), can be factored into the product p(A|C) times p(B|C) for all values of C if A and B are conditionally independent of C. The authors should argue, or preferably demonstrate, that the necessary conditional independence properties approximately hold for the application of their method to climate model fields. The feedbacks across scales in the climate system and the coupled nature of the physical equations may serve as a basis for some degree of conditional independence, though I expect some cases where p(A|C) is a poor approximation of p(A|B,C) as implied by conditional independence.

Item 2. The authors appear to neglect temporal relationships in the method and example, even though such relationships are prevalent in the climate system. A perturbation in the pattern of sea surface temperature in the tropics, for example, may take months before the signal shows up in the spatial distribution of precipitation in the mid-latitudes. While introducing temporal correlations into their method is beyond the scope of the manuscript and not required at this stage, it would still be beneficial to readers if the authors described how their method could be extended is this way.

Item 3. The opening paragraph states that there is skepticism in using a scalar metric to assess climate model performance. This gives the impression that everything gets boiled down to a single number, which isn't the case. Climate models are often assessed using a vector of scalar quantities (e.g. as in Gleckler et al), a scalar measure of a vector field, or combinations of these and other metrics. A single field projected on a Taylor diagram, for example, considers two orthogonal scalar quantities (centered rms and correlation). Please clarify the description.

Item 4. The opening paragraph also describes the need to account for spatial and field dependencies. Field dependence is an essential feature of your methodology, so

it would be useful to readers to provide a specific example of what you mean by field dependence early in the introduction.

Item 5. The first sentence in the second paragraph in the introduction is a little awkward and should be rewritten (lines 12-15, page 3). There is an observational record of climate, but not an observational record of a climate model. Moreover, this statement seems to suggest that data assimilation is primarily a data imputation method, which it really isn't. Data assimilation minimizes the differences between the model state and observations, while insuring that the state fields abide by conservation laws (mass, energy, and momentum) and other important physical dependencies. This paragraph overall seems to imply that the models don't do a very good job with the dependencies, which is arguable. I have confidence that the models are getting many of the large scale dependencies about right (e.g. equator to pole gradients, land-ocean contrasts, temperature dependence of water vapor through Clausius-Clapeyron, etc).

Item 6. It's a good idea to present the general idea behind the metric in equation (1) in the introduction, though I found myself flipping back and forth between the introduction and section 2 to make better sense of the information. To make it easier for readers to get through the introduction without getting hung up on details, perhaps you could introduce the concept in more general terms. Also, the symbol Z is used for the metric in this section, but it doesn't appear elsewhere in the manuscript and should be dropped. And the times symbol in 'n_obs x n_pts' on line 3, page 4 suggests that v is a matrix with n_obs rows and n_pts columns. I recommend changing it to n_obs n_pts.

Item 7. There is a typo in the lower right element of the S inverse matrix on line 11, page 7. The sigma index should be 22, instead of 11. For consistency, use the same indices for off-diagonal terms (e.g. S12 is used for the lower left term on line 11, while S21 is used on line 13 page 7).

Item 8. Regarding the alpha parameter, please provide references or further information about the statement that alpha depends only on the geometry of the neighborhood

and not on the details of the fields. I have other questions about alpha. How much does it vary going from a first order neighborhood to a higher order neighborhood? Can alpha be extended from a scalar to a vector to optimize the covariances in different regions in the neighborhood?

Item 9. The witch hat plots are convenient, but take some effort to get used to. It would be useful it you first stepped the readers through the concept with a simple example. How much does the shape of the witch hat depend on the selected indexing for the neighborhood? E.g. swapping x3 and x4 in figure 1 appears arbitrary, but results in a different Q. Does the averaging of the cells for a given distance from the diagonal hide information that could be important? Are there other simple ways to show the differences between the empirical and GMRF estimates (e.g. Hinton diagrams)?

Item 10. Figures 3 and 4 are positioned before section 4 in the manuscript, but the figures rely on information about the climate model data from that section (e.g. the estimates are from 15 samples). Please cross reference the material from section 4 where needed to avoid confusion.

Item 11. In the last paragraph on page 11, the authors state that the only meaningful correlations are of TREFHT with PSL and PRECT with PSL. However, if TREFHT and PRECT are individually correlated with PSL, shouldn't TREFHT and PRECT also be correlated to each other? Moreover, there is a contradiction between the physical explanation on lines 23-25, page 11 and the sign of the correlation between PSL and PRECT (low pressure systems increase precipitation).

Item 12. The model simulations use prescribed sea surface temperatures, which strongly constrain the near surface air temperature, so it seems surprising that the biggest changes in cost are associated with the 2-m air temperature. Can the authors provide a physical explanation for their finding?

Item 13. The authors find that spatial dependencies are more important to capture than field dependencies for the four selected outputs (PSL, TREFHT, U, PRECT). Do

they have any reason to suspect (or can they show) that the field dependencies will dominate over spatial dependencies for other fields? If not, then this suggests that it may not be critical to capture the field dependencies and that their method does not offer many clear benefits over standard model assessment techniques. From their example in figure 5 of optimizing model performance by changing two parameters (c0 and Ke), it even looks like adding the spatial dependence alone would not greatly affect the conclusions drawn from assuming spatial independence (i.e., that high values of c0 and low values of Ke are best).

Item 14. Observational uncertainty does not appear to be taken into account in their method. Can the authors comment on and suggest ways to incorporate observational uncertainty into their test statistic?
* * *

---

## Referee Comment (RC2) · Anonymous Referee #2 · 11 Mar 2016

This study computes multivariate cost functions which can be used to choose optimal values of climate-model tuning parameters. These cost functions are special because they take correlations between neighboring points and between the variables analyzed into account. Using these special cost functions is shown to make little difference when optimizing 2 cloud parameters in CAM3.1 for 2m T, 200mb zonal winds, surface pressure, and surface precipitation. On one hand, lack of sensitivity implies the methodology is probably implemented reasonably well. On the other hand, lack of sensitivity begs the question of whether the sophistication is necessary.

Correlations in space and across variables were handled by Gaussian Markov Random Fields (GMRFs). I had a hard time understanding whether this is an appropriate

technique or whether it was implemented correctly. In particular:

1. Equation 1 is introduced in the introduction and is said to be the culmination of the subsequent derivations but is never fully explained. Better explanation is needed. In particular, I don't think it makes sense to provide this equation in the introduction.

2. I think Eq. 1 is a log-likelihood function derived from assuming model errors follow a multivariate Gaussian distribution (eq. 2) with the inverse covariance matrix $\Sigma^{-1}$ replaced by GMRF precision matrix. These points need clarification and the reasonableness of assuming a multivariate Gaussian distribution for model output and for approximating the covariance matrix with a GMRF precision matrix both require further justification.

3. I think the log-likelihood function in eq 1 is missing the following term: $ln((2\pi)^{-}n/2\text{tr}(\Sigma)^{-1/2})$. Is this true? In any case, this derivation needs to be more clear.

4. The precision matrix is only described for the 2x2 case. Are the "rules" on page 6 followed only for the 2x2 case, or are they followed for all cases?

5. Because Q seems to be defined independently of the spatial autocorrelation in the actual data, I find it hard to believe that it can be a good approximation for $\Sigma^{-1}$ except by chance. In particular, I bet the "witch hat graph" for surface precipitation alone (which has short autocorrelation lengthscales) looks very different than that for surface temperature (which has long autocorrelation lengthscales) and that Fig. 4 only looks reasonable for quantities which happen to have the autocorrelation structure matching the precision matrix assumptions. I would like to see the comparison between $Q^{-1}$ and $\Sigma$ (note I'm asking for things in correlation-matrix space rather than precision matrix space because the former is easier to interpret physically) for several different output fields to gain confidence in the method. The fact that Q is defined independently of autocorrelation in the actual data is my single biggest concern with this paper.

[Figure]

5. The fact that the precision matrix has a zero eigenvalue seems to be an obvious result of the fact that Q indicates the neighbors of each cell and neighbors of the last cell can be predicted from the others (because the cells which are its neighbors have already tagged it as being their neighbor). I am surprised and alarmed that your solution to this problem is to add a small perturbation to make your singular matrix merely nearly-singular. It seems like this nearly-singular matrix will at best have numerical issues and at worst isn't actually solving the system you meant to solve. Wouldn't it make more sense to replace the system with a matrix of 1 lower dimension?

*Other Comments:*

1. In the title and elsewhere, you call your method a "metric". I think the benefit of your approach is that it allows you to evaluate a log-likelihood function in order to choose the best parameter settings fo an uncertainty-quantification problem. While the log-likelihood function does give you a scalar value for a particular set of parameters and is therefore a metric of sorts, I think emphasizing that you're defining a metric is kind of missing the main point of what you're doing. In particular, you have to specify exactly what output you want to use to define a metric and I think a benfit of your method is that it should work on a wide variety of output data choices. In short, I'd suggest changing "metric" to "method" throughout the text.

2. using CAM3.1 is odd and detracts from the publication-worthiness of the paper because it is an ancient model which nobody cares about anymore. Can you really not find data from more recent model runs? It would be worth the effort.

3. In eq. 2, you need to indicate that |x| is the determinant of x.

4. p. 6 line 8: "fuller" should be "more full"

5. You should define what the Kronecker product is for climate people, who may not know off the top of their heads.

6. p. 7, line 18: you're missing a word between supplemental and carries.

7. p. 11 line 25: low pressure cooling the underlying surface would be a *positive* correlation. Perhaps you're seeing a "thermal low" effect?

8. You show in Fig. 5 that using GMDV or not doesn't make a big difference. Is this the result of your particular choice of parameters and/or model version and/or output variables? Taking the time to test your method in other cases would at a minimum make your conclusions more robust and could potentially show that your method has an important impact in certain circumstances.

9. In the supplementary material, why assume x has means which are all zero?

*Recommendation*:

In summary, I don't have confidence based on the information conveyed in the current draft that this approach makes sense. Part of this is that the methodology is not clearly explained (which can be fixed by rewriting). The other part is that the method seems to assume a spatial autocorrelation structure without any consideration of the covariance structure of the actual data. For these reasons I think the paper should be rejected with encouragement to resubmit once these problems are fixed. Another problem with this paper is that its results are not particularly exciting - using the complicated methodology which is the core of the paper has little impact on model results. I don't think this is reason to reject the paper (because negative results are important to present so others don't repeat them), but is reason to be less enthusiastic in encouraging a resubmit.

---

## Author Comment (AC1) · 9 Apr 2016

*Response to reviewer # 1 and #2 input on* "A new metric for climate models that includes field and spatial dependencies using Gaussian Markov Random Fields" by Nosedal-Sanchez et al.

April 9, 2016

**Response to Reviewer # 1** Reviewer comment given in blue.

1. Conditional independence is an assumption underlying Markov random fields. For three variables A, B, and C, the joint probability distribution of A and B conditioned on C, written as $p(A,B|C)$, can be factored into the product $p(A|C)$ times $p(B|C)$ for all values of C if A and B are conditionally independent of C. The authors should argue, or preferably demonstrate, that the necessary conditional independence properties approximately hold for the application of their method to climate model fields. The feedbacks across scales in the climate system and the coupled nature of the physical equations may serve as a basis for some degree of conditional independence, though I expect some cases where $p(A|C)$ is a poor approximation of $p(A|B,C)$ as implied by conditional independence.

   The assumption of conditional independence (for estimating precisions among points outside a neighborhood structure) does not need to be met exactly in order for GMRF to represent a useful step forward toward the goals outlined in the introduction. The assumption facilitates the sparse representation of the precision matrix and therefore is convenient. It enables us to capture some but perhaps not the full extent of the dependencies that exist across space and fields in the climate data. The witch-hat graphs provide a measure of how well GMRF captures observed covariances. One could enlarge the neighborhood structure indicating conditional dependencies of the precision matrix beyond nearest neighbors, but we felt that the present treatment was adequate.

2. The authors appear to neglect temporal relationships in the method and example, even though such relationships are prevalent in the climate system. A perturbation

in the pattern of sea surface temperature in the tropics, for example, may take months before the signal shows up in the spatial distribution of precipitation in the mid-latitudes. While introducing temporal correlations into their method is beyond the scope of the manuscript and not required at this stage, it would still be beneficial to readers if the authors described how their method could be extended is this way.

It is common for climate model evaluation to place most of its emphasis on long-term means and that is our target application. GMRFs may be extended to include temporal relationships (e.g. Cressie and Wikle, 2011), but we did not attempt to develop those ideas in the present manuscript. Note that the assumption that the distribution of errors are Gaussian does not hold as well on short(er) time scales. The text mentions that the effects of teleconnection patterns shape local covariances (through a set of processes that are influenced by anomalously low or high pressures). Space-field GMRFs would be sensitive to these effects since the teleconnections shape long term means. We will update the text to provide more information about the possibility to extend the analysis to include temporal relationships.

Statistics for Spatio-Temporal Data, by Noel Cressie and Christopher K. Wikle. Wiley, Hoboken, NJ, 2011 (588 pp.)

3. The opening paragraph states that there is skepticism in using a scalar metric to assess climate model performance. This gives the impression that everything gets boiled down to a single number, which isn't the case. Climate models are often assessed using a vector of scalar quantities (e.g. as in Gleckler et al), a scalar measure of a vector field, or combinations of these and other metrics. A single field projected on a Taylor diagram, for example, considers two orthogonal scalar quantities (centered rms and correlation). Please clarify the description.

We agree with the reviewer's point that the scientific community makes use of many metrics to judge a model's credibility. The section explaining our point was poorly written. The issue is not that climate scientists already make use of many metrics in model selection. We needed to first say that formal methods for model calibration operate on a single scalar metric. The scientific community is skeptical that a scalar metric (and therefore formal calibration methods) could adequately capture all the scientific sensibilities that are needed for judging model acceptability. We will clarify this point.

4. The opening paragraph also describes the need to account for spatial and field dependencies. Field dependence is an essential feature of your methodology, so it would be useful to readers to provide a specific example of what you mean by field dependence early in the introduction.

Thanks for this suggestion. We will provide an example.

5. The first sentence in the second paragraph in the introduction is a little awkward and should be rewritten (lines 12-15, page 3). There is an observational record of climate, but not an observational record of a climate model. Moreover, this statement seems to suggest that data assimilation is primarily a data imputation method, which it really isn't. Data assimilation minimizes the differences between the model state and observations, while insuring that the state fields abide by conservation laws (mass, energy, and momentum) and other important physical dependencies. This paragraph overall seems to imply that the models don't do a very good job with the dependencies, which is arguable. I have confidence that the models are getting many of the large scale dependencies about right (e.g. equator to pole gradients, land-ocean contrasts, temperature dependence of water vapor through Clausius-Clapeyron, etc).

In this paragraph we wish to make the point that there exists a very limited observational record on which to estimate space and field dependencies of climate phenomena. Likely the best synthesis of these dependencies are from reanalysis products for some of the reasons you state. However there is a catch. The models used for data assimilation rely on their own physics packages to predict cloud characteristics and their radiative effects. Moreover, the data assimilation strategies for generating these products do not attempt to conserve mass, energy, and momentum, particularly between analysis steps. So the products are both a reflection of the observations that go into them and the physics and fluid motions of the model. Figure 1 below provides an illustration of this point. It shows that a multivariate measure of the distance between NCEP and ERA40, which are two reanalysis products, and CAM3.1 were nearly as different from one another as CAM3.1 was to them. Seasonally and regionally, the two products contained upwards of 200 $\mathrm{Wm}^{-2}$ differences in shortwave radiation reaching the surface which is emblematic of the different parameterizations each model uses for estimating cloud distributions and their radiative properties.

6. It's a good idea to present the general idea behind the metric in equation (1) in the introduction, though I found myself flipping back and forth between the introduction and section 2 to make better sense of the information. To make it easier for readers to get through the introduction without getting hung up on details, perhaps you could introduce the concept in more general terms. Also, the symbol Z is used for the metric in this section, but it doesn't appear elsewhere in the manuscript and should be dropped. And the times symbol in '$n_{obs}$ x $n_{pts}$' on line 3, page 4 suggests that v is a matrix with $n_{obs}$ rows and $n_{pts}$ columns. I recommend changing it to $n_{obs}n_{pts}$.

[Figure]

**Figure 1:** Average distance between two data assimilation products, NCEP (Kalnay et al.,1996); Kistler et al. 2001) and ERA40 (Uppala et al. 2005), is a good fraction of the distance to CAM3.1 (Community Atmosphere Model version 3.1). The length of each segment is based on the metric used in Jackson et al., (2008) and described by Mu et al., (2003) and includes shortwave radiation to surface, 2 m air temperature, surface sensible heat flux, relative humidity, air temperature, zonal winds, and sea level pressure from 1990 to 2001.

These are very good suggestions and clarifications. We will update the text.

7. There is a typo in the lower right element of the S inverse matrix on line 11, page 7. The sigma index should be 22, instead of 11. For consistency, use the same indices for off-diagonal terms (e.g. S12 is used for the lower left term on line 11, while S21 is used on line 13 page 7).

Thanks for catching this error.

8. Regarding the alpha parameter, please provide references or further information about the statement that alpha depends only on the geometry of the neighborhood and not on the details of the fields. I have other questions about alpha. How much does it vary going from a first order neighborhood to a higher order neighborhood? Can alpha be extended from a scalar to a vector to optimize the covariances in

Equation (5) shows that $\alpha$ is only dependent on the eigenvalues of the Q matrix. The Q matrix itself is only a function of the domain (e.g. geometry and number of latitude and longitude grid points) and the neighborhood structure. Thus we would expect the value for $\alpha$ to be affected if we use a higher order neighborhood. We started the task of building a Q matrix with a higher order neighborhood structure, however implementing it correctly requires a lot of attention to detail to deal with how the stencil changes as one approaches a boundary and this task will require more time to complete than we have at the moment. Thus we don't know how much $\alpha$ would be affected. Because the higher order Q matrix will have more than one singular vector, we came to the realization that the concept of $\alpha$ may need to be expanded to accommodate all of the Q matrix singular vectors. Thus the answer to the question is not straight-forward and would require further consideration. In response to your last question, since $\alpha$ in the way we have been using it exists as an extension of the Q matrix, it does not make sense to use it to accommodate covariances for particular regions which may be field dependent.

9. The witch hat plots are convenient, but take some effort to get used to. It would be useful it you first stepped the readers through the concept with a simple example. How much does the shape of the witch hat depend on the selected indexing for the neighborhood? E.g. swapping x3 and x4 in figure 1 appears arbitrary, but results in a different Q. Does the averaging of the cells for a given distance from the diagonal hide information that could be important? Are there other simple ways to show the differences between the empirical and GMRF estimates (e.g. Hinton diagrams)?

The results would not be altered by how boxes are indexed within a Q matrix that correctly identifies the neighborhood structure around each grid cell. The reason to present a summary of the covariance matrix in terms of a 'witch hat' graph is because there is not much variation in the estimate of the variances/covariances along any of the diagonals. There can be small deviations in the symmetry that occur because of how neighboring cells are indexed particularly as one approaches a boundary. An example of this deviation is provided below. However a fairly accurate estimate of the 'witch hat' could be constructed as an average of variances or covariances along the various diagonals relative to the main diagonal. However, as the example illustrates, this is not as simple as an evaluation of the distance to the diagonal. We construct our covariances for the 'witch hat' graph by explicitly identifying those cells that are a given distance from the diagonal (see example below). Hinton diagrams provide a graphic view of the size of values within a matrix. 'witch hat' graphs allow us to compare GMRF implied variances/covariances with those estimated empirically from data. The text explaining witch hats will be

further clarified.

**Example of the construction of a 'witch hat'.**

We will describe the construction of a witch hat graph for a $3 \times 3$ lattice, like the one shown below.

| 1 | 2 | 3 |
|---|---|---|
| 4 | 5 | 6 |
| 7 | 8 | 9 |

Suppose that variance estimates are available at each of the 9 grid points for one field, for example $S_{11}Q^*$. In this case, $S_{11}Q^*$ is a $9 \times 9$ matrix.

Like any other graph, a witch hat graph is formed by points. We will find the points that define a witch hat graph that makes comparisons in the N - S direction. From the figure shown above, it is clear that grid cells $1, 2, 3, 4, 5$, and 6, have one grid cell below them: 4, 5, 6, 7, 8, and 9, respectively. Now, we use these numbers to form pairs: $(4,1), (5,2), (6,3), (7,4), (8,5)$, and $(9,6)$. Then, using the corresponding elements of our matrix of estimates, we compute the following average:

$$w(-1) = \frac{\hat{\sigma}_{41} + \hat{\sigma}_{52} + \hat{\sigma}_{63} + \hat{\sigma}_{74} + \hat{\sigma}_{85} + \hat{\sigma}_{96}}{6}$$

(where $\hat{\sigma}_{ij} = \hat{\sigma}(i,j)$ = element located on $i$th row and $j$th column of matrix of estimates).

Thus, we define $(-1, w(-1))$ as one point of our witch hat graph. Let us find another point. Again, using the same figure, it is clear that grid cells $4, 5, 6, 7, 8$, and 9, have one grid cell above them: $1, 2, 3, 4, 5$, and 6, respectively. As we did before, we proceed to form pairs with these numbers: $(1,4), (2,5), (3,6), (4,7), (5,8)$, and $(6,9)$. Then, we use the corresponding elements of our matrix of estimates to compute another average:

$$w(1) = \frac{\hat{\sigma}_{14} + \hat{\sigma}_{25} + \hat{\sigma}_{36} + \hat{\sigma}_{47} + \hat{\sigma}_{58} + \hat{\sigma}_{69}}{6}$$

This couple of numbers, $(1, w(1))$, gives another point of the witch hat graph. Doing something similar, with grid cells that have neighbours located two rows down or up of themselves, we obtain:

$$w(-2) = \frac{\hat{\sigma}_{71} + \hat{\sigma}_{82} + \hat{\sigma}_{93}}{3}$$

and

$$w(2) = \frac{\hat{\sigma}_{17} + \hat{\sigma}_{28} + \hat{\sigma}_{39}}{3}.$$

Note that the number of cells from one grid to itself is zero. So, a fifth point for our graph is

$$w(0) = \frac{\hat{\sigma}_{11} + \hat{\sigma}_{22} + \ldots + \hat{\sigma}_{99}}{9}.$$

A witch hat graph is a graphical representation of these pairs of points: $(-2, w(-2))$, $(-1, w(-1))$, $(0, w(0))$, $(1, w(1))$, and $(2, w(2))$. By construction, $w(-1) = w(1)$ and $w(-2) = w(2)$ (recalling that the matrix of estimates is symmetric). Which is convenient for computational purposes. It is worth noting that we could define a graph in the E-W direction in a similar fashion. In general, a witch hat graph in the E-W direction will differ from the one constructed to make comparisons in the N-S direction.

Note. Making a graph of a $9 \times 9$ matrix of estimates would allow us to see that $w(1)$ = average of entries located **three** columns to the right of main diagonal. Similarly, $w(2)$ = average of entries located **six** columns to the right of main diagonal. However, these numbers (three and six) **depend on the number of columns of lattice in question**.

| $\hat{\sigma}_{11}$ | $\hat{\sigma}_{12}$ | $\hat{\sigma}_{13}$ | $\hat{\sigma}_{14}$ | $\hat{\sigma}_{15}$ | $\hat{\sigma}_{16}$ | $\hat{\sigma}_{17}$ | $\hat{\sigma}_{18}$ | $\hat{\sigma}_{19}$ |
|---|---|---|---|---|---|---|---|---|
| $\hat{\sigma}_{21}$ | $\hat{\sigma}_{22}$ | $\hat{\sigma}_{23}$ | $\hat{\sigma}_{24}$ | $\hat{\sigma}_{25}$ | $\hat{\sigma}_{26}$ | $\hat{\sigma}_{27}$ | $\hat{\sigma}_{28}$ | $\hat{\sigma}_{29}$ |
| $\hat{\sigma}_{31}$ | $\hat{\sigma}_{32}$ | $\hat{\sigma}_{33}$ | $\hat{\sigma}_{34}$ | $\hat{\sigma}_{35}$ | $\hat{\sigma}_{36}$ | $\hat{\sigma}_{37}$ | $\hat{\sigma}_{38}$ | $\hat{\sigma}_{39}$ |
| $\hat{\sigma}_{41}$ | $\hat{\sigma}_{42}$ | $\hat{\sigma}_{43}$ | $\hat{\sigma}_{44}$ | $\hat{\sigma}_{45}$ | $\hat{\sigma}_{46}$ | $\hat{\sigma}_{47}$ | $\hat{\sigma}_{48}$ | $\hat{\sigma}_{49}$ |
| $\hat{\sigma}_{51}$ | $\hat{\sigma}_{52}$ | $\hat{\sigma}_{53}$ | $\hat{\sigma}_{54}$ | $\hat{\sigma}_{55}$ | $\hat{\sigma}_{56}$ | $\hat{\sigma}_{57}$ | $\hat{\sigma}_{58}$ | $\hat{\sigma}_{59}$ |
| $\hat{\sigma}_{61}$ | $\hat{\sigma}_{62}$ | $\hat{\sigma}_{63}$ | $\hat{\sigma}_{64}$ | $\hat{\sigma}_{65}$ | $\hat{\sigma}_{66}$ | $\hat{\sigma}_{67}$ | $\hat{\sigma}_{68}$ | $\hat{\sigma}_{69}$ |
| $\hat{\sigma}_{71}$ | $\hat{\sigma}_{72}$ | $\hat{\sigma}_{73}$ | $\hat{\sigma}_{74}$ | $\hat{\sigma}_{75}$ | $\hat{\sigma}_{76}$ | $\hat{\sigma}_{77}$ | $\hat{\sigma}_{78}$ | $\hat{\sigma}_{79}$ |
| $\hat{\sigma}_{81}$ | $\hat{\sigma}_{82}$ | $\hat{\sigma}_{83}$ | $\hat{\sigma}_{84}$ | $\hat{\sigma}_{85}$ | $\hat{\sigma}_{86}$ | $\hat{\sigma}_{87}$ | $\hat{\sigma}_{88}$ | $\hat{\sigma}_{89}$ |
| $\hat{\sigma}_{91}$ | $\hat{\sigma}_{92}$ | $\hat{\sigma}_{93}$ | $\hat{\sigma}_{94}$ | $\hat{\sigma}_{95}$ | $\hat{\sigma}_{96}$ | $\hat{\sigma}_{97}$ | $\hat{\sigma}_{98}$ | $\hat{\sigma}_{99}$ |

Estimates in blue represent numbers that would be used to make graph in the E-W direction. Estimates in red represent numbers that would be used to make graph in the S-N direction. This suggests that using the "second main diagonal" to plot witch hat graphs at $w(-1) = w(1)$ will result in a very different value.

10. Figures 3 and 4 are positioned before section 4 in the manuscript, but the figures rely on information about the climate model data from that section (e.g. the estimates are from 15 samples). Please cross reference the material from section 4 where needed to avoid confusion.

Thank you for this suggestion.

11. In the last paragraph on page 11, the authors state that the only meaningful correlations are of TREFHT with PSL and PRECT with PSL. However, if TREFHT and PRECT are individually correlated with PSL, shouldn't TREFHT and PRECT also be correlated to each other? Moreover, there is a contradiction between the physical explanation on lines 23-25, page 11 and the sign of the correlation between PSL and PRECT (low pressure systems increase precipitation).

To address this question we created maps of the grid point correlations between JJA mean 2m air temperature (TREFHT), sea level pressure (PSL), and precipitation (PRECT) with sea level pressure (PSL) (Figure 2). What is clear between all these figures is that there is a lot of structure to all these maps. The sign of the correlation is regionally dependent. Therefore providing a mechanistic explanation of the spatially averaged correlation is not going to be particularly meaningful. However it may be useful for readers to know that there is a lot of structure to these maps and that the reason that the spatially averaged correlation between PSL and PRECT is so small is not because local correlations are small. Rather the average includes regions of large negative correlations as well as regions with large positive correlations. Despite losing this regional information in the S matrix summary of field covariances, this does not affect GMRF estimated field covariances between these fields as can be seen within the 'witch hat' graphs.

12. The model simulations use prescribed sea surface temperatures, which strongly constrain the near surface air temperature, so it seems surprising that the biggest changes in cost are associated with the 2-m air temperature. Can the authors provide a physical explanation for their finding?

By our read of manuscript Figure 6, cost changes related to 2m air temperature (TREFHT) are the smallest relative to the three other fields. It is true that specifying sea surface temperatures will limit the model's TREFHT response over the ocean, however the models response to changes in parameters can affect the atmospheric boundary layer over the ocean including TREFHT. Moreover TREFHT is less restricted over land which was an important fact explaining why TREFHT showed the biggest qualitative differences in cost when using the Q matrix to include space dependencies within the cost. The latter is due to the sensitivity of the Q operator to the sharp spatial structures that arise from model-observational differences in and around the poorly resolved Andes mountains.

13. The authors find that spatial dependencies are more important to capture than field dependencies for the four selected outputs (PSL, TREFHT, U, PRECT). Do they have any reason to suspect (or can they show) that the field dependencies will dominate over spatial dependencies for other fields? If not, then this suggests that it may not be critical to capture the field dependencies and that their method does not offer many clear benefits over standard model assessment techniques. From their example in figure 5 of optimizing model performance by changing two parameters (c0 and Ke), it even looks like adding the spatial dependence alone would not greatly affect the conclusions drawn from assuming spatial independence (i.e., that high values of c0 and low values of Ke are best).

    While the current results do not provide a strong case for why including field dependencies is important, we were only looking at four fields within the tropics in JJA. We don't yet know whether field dependencies become important for other fields, regions, or seasons. We elected to keep the scope of the present manuscript focused on the mathematical treatment of GMRF. It is helpful to know that the results look reasonable which would be hard to evaluate without this limited example. We are in the process of generating results for 11 fields, 3 regions, and 4 seasons which will be reported separately. It also could be that the importance of field dependencies may depend on what parameters are being varied.

14. Observational uncertainty does not appear to be taken into account in their method. Can the authors comment on and suggest ways to incorporate observational uncertainty into their test statistic?

    This is an excellent and important question. The most obvious place to include this information is within the S matrix. For instance if a grid point error variance is known, it could be added to the diagonal elements. However we have already run into a case where satellite observations of cloud fields include linear structures that are obviously related to the satellite tracks. We suspect that the Q operator within GMRF may be particularly sensitive to these artifacts in the data. We do not have a full answer to this question. There is not much experience in the community as a whole for quantifying these uncertainties and representing them within metrics of climate model performance.

    **Response to Reviewer # 2** Reviewer comment given in blue.
    Correlations in space and across variables were handled by Gaussian Markov Random Fields (GMRFs). I had a hard time understanding whether this is an appropriate technique or whether it was implemented correctly. In particular:

 1. Equation 1 is introduced in the introduction and is said to be the culmination of the subsequent derivations but is never fully explained. Better explanation is needed. In particular, I dont think it makes sense to provide this equation in the introduction.

We will remove the specific details about GMRF, including equation (1) to a subsequent section.

2. I think Eq. 1 is a log-likelihood function derived from assuming model errors follow a multivariate Gaussian distribution (eq. 2) with the inverse covariance matrix $\Sigma^{-1}$ replaced by GMRF precision matrix. These points need clarification and the reasonableness of assuming a multivariate Gaussian distribution for model output and for approximating the covariance matrix with a GMRF precision matrix both require further justification.

Currently climate model evaluation makes use of relatively short, few year model integrations for testing the effects of uncertain parameters. When compared to the 10 to 30-year climatologies of observations the distribution of errors is approximately Gaussian (see Figure 3). The distribution of climate fields on very short, hourly to daily time scales can be decidedly non-Gaussian which is not the case for longer term means. We include here a few examples of monthly mean climate model output that show an approximately Gaussian distribution. The reasonableness for using GMRF to estimate the inverse covariance matrix is provided by the 'witch-hat' graphs.

3. I think the log-likelihood function in eq 1 is missing the following term: $ln((2\pi)^{-n/2}tr(\Sigma)^{-1/2})$. Is this true? In any case, this derivation needs to be more clear.

Yes this is true, although it would include the determinant of the covariance matrix not its trace. Within the statistics community the argument of the likelihood function is referred to as the log-likelihood since the factor you are referring to is a constant offset. However this can be confusing especially since this community often also neglects the factor of $\frac{1}{2}$ that should be included within the exponential argument for a Gaussian distribution. We will make our statement more clear.

4. The precision matrix is only described for the 2x2 case. Are the "rules" on page 6 followed only for the 2x2 case, or are they followed for all cases?

Yes, the rules apply to all cases.

5. Because Q seems to be defined independently of the spatial autocorrelation in the actual data, I find it hard to believe that it can be a good approximation for $\Sigma^{-1}$ except by chance. In particular, I bet the "witch hat graph" for surface precipitation alone (which has short autocorrelation length scales) looks very different than that for surface temperature (which has long autocorrelation length scales) and that Fig. 4 only looks reasonable for quantities which happen to have the autocorrelation structure matching the precision matrix assumptions. I would like to see the comparison between $Q^{-1}$ and $\Sigma$ (note Im asking for things in correlation-matrix

space rather than precision matrix space because the former is easier to interpret physically) for several different output fields to gain confidence in the method. The fact that Q is defined independently of autocorrelation in the actual data is my single biggest concern with this paper.

You are correct that the Q matrix is defined independently of field information. This matrix is a differential operator which 'senses' how much fields change within the neighborhood structure. The units come from scaling the Q matrix with $S^{-1}$ using the Kronecker product. Together the GMRF provides a decent approximation to the inverse covariance matrix $\Sigma^{-1}$. 'Witch-hat' graphs show observed variances and covariances with the inverse precisions (implied variances/covariances) and are being used to test how well GMRF capture observed space and field dependencies. Figures 4 and 5 show several additional 'witch-hat' graphs to provide a more complete evaluation of GMRFs representation of observed space and field dependencies.

6. The fact that the precision matrix has a zero eigenvalue seems to be an obvious result of the fact that Q indicates the neighbors of each cell and neighbors of the last cell can be predicted from the others (because the cells which are its neighbors have already tagged it as being their neighbor). I am surprised and alarmed that your solution to this problem is to add a small perturbation to make your singular matrix merely nearly-singular. It seems like this nearly-singular matrix will at best have numerical issues and at worst isn't actually solving the system you meant to solve. Wouldnt it make more sense to replace the system with a matrix of 1 lower dimension?

The issue is that Q matrix needs to have the same dimension as each field so changing its dimension is not the solution. Note that we only need to take an inverse of the Q matrix for testing GMRF predictions of observed covariances within the 'witch-hat' graphs. Even then the R codes provided robust results.

*Other Comments:*

1. In the title and elsewhere, you call your method a "metric". I think the benefit of your approach is that it allows you to evaluate a log-likelihood function in order to choose the best parameter settings to an uncertainty-quantification problem. While the log-likelihood function does give you a scalar value for a particular set of parameters and is therefore a metric of sorts, I think emphasizing that youre defining a metric is kind of missing the main point of what you're doing. In particular, you have to specify exactly what output you want to use to define a metric and I think a benefit of your method is that it should work on a wide variety of output data choices. In short, Id suggest changing "metric" to "method" throughout the text.

We appreciate your thoughts on this matter and agree with your point. The term "metric" is often used for climate model evaluation although that term does not capture the fact that we are evaluating a signal to noise ratio for testing the null hypothesis of whether changes in a climate model are significant. In other publications we sometimes refer to this normalized metric as a "test statistic". We therefor prefer using that term over "method". We thank you for this suggestion.

2. using CAM3.1 is odd and detracts from the publication-worthiness of the paper because it is an ancient model which nobody cares about anymore. Can you really not find data from more recent model runs? It would be worth the effort.

   CAM3.1 output is more than adequate for purposes of examining how GMRF would be applied to climate model evaluation.

3. In eq. 2, you need to indicate that $|x|$ is the determinant of x.

   We can indicate this.

4. p. 6 line 8: "fuller" should be "more full"

   Thanks for catching this.

5. You should define what the Kronecker product is for climate people, who may not know off the top of their heads.

   We can provide a the following example of how the Kronecker product works: Consider the following $2 \times 2$ matrices

$$A = \begin{pmatrix} 1 & 4 \\ 2 & 5 \end{pmatrix} \text{ and } B = \begin{pmatrix} 1 & 3 \\ 0 & 4 \end{pmatrix}.$$

   The Kronecker product of $A$ and $B$, $A \otimes B$, is given by

$$A \otimes B = \begin{pmatrix} 1(B) & 4(B) \\ 2(B) & 5(B) \end{pmatrix} = \begin{pmatrix} 1 & 3 & 4 & 12 \\ 0 & 4 & 0 & 16 \\ 2 & 6 & 5 & 15 \\ 0 & 8 & 0 & 20 \end{pmatrix}$$

6. p. 7, line 18: youre missing a word between supplemental and carries.

   Thanks for catching this.

7. p. 11 line 25: low pressure cooling the underlying surface would be a *positive* correlation. Perhaps youre seeing a "thermal low" effect?

See also response to item 12 from reviewer # 1. Because covariances vary by region (e.g. Figure 2), we will back off from providing particular explanations for explaining covariances that may only apply to particular regions of the domain.

8. You show in Fig. 5 that using GMRF or not doesnt make a big difference. Is this the result of your particular choice of parameters and/or model version and/or output variables? Taking the time to test your method in other cases would at a minimum make your conclusions more robust and could potentially show that your method has an important impact in certain circumstances.

While we agree that testing GMRF in all possible cases (with more fields, regions, seasons, and model parameters) would provide a more thorough examination of the question you raise of whether the GMRF can make an important difference, our purpose for the present manuscript was to develop the mathematical application of GMRF to climate model output. The effort represents several years of concerted effort. The testing of GMRF in more cases is being developed with more scientific goals in mind.

9. In the supplementary material, why assume x has means which are all zero?

It was not a necessary assumption, but it did facilitate the derivation without complicating the expressions.

[Figure]

**Figure 2:** JJA correlations between 2m air temperature (TREFHT), sea level pressure (PSL), and precipitation (PRECT).

[Figure]

**Figure 3:** Histograms of differences between observations and model output for four fields (U, TREFHT, PSL, and PRECT) for an experiment that includes changes to cloud parameters C0 and ke.

[Figure]

**Figure 4:** 'Witch hat' graphs testing GMRF approximations to empirical estimates of variances of U, PSL, TREFHT, and PRECT.

[Figure]

**Figure 5:** 'Witch hat' graphs testing GMRF approximations to empirical estimates of covariances between TREFHT and PRECT.